# Assisted Reproductive Technology in Neotropical Deer: A Model Approach to Preserving Genetic Diversity

**DOI:** 10.3390/ani11071961

**Published:** 2021-06-30

**Authors:** Luciana Diniz Rola, Marcos Eli Buzanskas, Luciana Magalhães Melo, Maiana Silva Chaves, Vicente José Figueirêdo Freitas, José Maurício Barbanti Duarte

**Affiliations:** 1Centro de Ciências Agrárias, Departamento de Zootecnia, Universidade Federal da Paraíba, Areia 58051-900, Brazil; luciana.diniz@academico.ufpb.br (L.D.R.); marcos@cca.ufpb.br (M.E.B.); 2Unidade de Pesquisa em Genética Molecular, Centro Universitário Fametro (UNIFAMETRO), Fortaleza 60010-260, Brazil; luciana.melo@professor.unifametro.edu.br; 3Laboratório de Fisiologia e Controle da Reprodução, Faculdade de Veterinária, Universidade Estadual do Ceará, Fortaleza 60714-903, Brazil; maiana.chaves@uece.br; 4Núcleo de Pesquisa e Conservação de Cervídeos (NUPECCE), Departamento de Zootecnia, Faculdade de Ciências Agrárias e Veterinárias, Universidade Estadual Paulista, Jaboticabal 01049-010, Brazil

**Keywords:** artificial insemination, Cervidae, cloning, conservation, embryo transfer, germplasm banks

## Abstract

**Simple Summary:**

Deer species in the Neotropical region have undergone a decline of their populations. Although conservation of their natural habitat is considered the best way to assist the species, the speed of occupation of these areas and the anthropic actions are so fast that the efforts are, at times, insufficient. As free-living populations decrease, there is a descent in the genetic diversity and an increase in crossbreeding between related individuals (inbreeding). Genetic diversity is essential for survival, since it enables natural selection to occur, providing adaptation and maintenance of the species. To protect the genetic diversity, it is possible to use reproductive techniques and conserve different types of cells, which can be used in the future to reestablish any alleles that have been lost by the populations.

**Abstract:**

One of the most significant challenges in deer is the ability to maintain genetic diversity, avoiding inbreeding and sustaining population health and reproduction. Although our general knowledge of reproductive physiology is improving, it appears that the application of assisted reproductive technology (ART) will more efficiently advance wildlife conservation efforts and preserve genetic diversity. The purpose of this review is to present the most important results obtained with the use of ART in Neotropical deer. Thus, the state-of-the-art for estrus synchronization, semen technology, artificial insemination, and in vivo embryo production will be presented. In vitro embryo production (IVP) is also a biotechnology that is taking initial steps in deer. In this aspect, the approach with the proteomics of ovarian follicular fluid is being used as a tool for a better understanding of oocyte maturation. Finally, cell banks and the use of interspecific somatic cell nuclear transfer (iSCNT) as well as the use of stem cells for gametes differentiation are promising techniques.

## 1. Introduction

The Neotropics contain several biodiversity hotspots and cover an impressive range of biomes, being considered one of the richest biogeographic regions in the world for deer diversity [1]. Among the seventeen species of Neotropical deer currently described, two categories are included: species of small and spike-like antlers, weighing less than 25 kg, adapted to forests and closed vegetation (genera *Mazama* and *Pudu*), and species with branched antlers, weighing more than 25 kg, inhabit open-field environments (genera *Odocoileus*, *Hippocamelus*, *Ozotoceros,* and *Blastocerus*) [2,3,4].

Biomes have faced high rates of habitat loss, and the habitat conversion for agriculture and other human activities have caused changes that affect many of these Neotropical deer species’ natural habitats. In addition, other anthropic actions are threatening the natural populations, such as overhunting, the introduction of new diseases, competition with production animals, construction of roads and hydroelectric plants, attacks by domestic dogs, among others [5,6]. These pressures imposed on species have caused population bottlenecks and segregation between populations, leading to difficulties in gene flow, decreased genetic diversity, and increased mating between related individuals (i.e., inbreeding) [7].

The maintenance of genetic diversity within populations, which are large enough to be sustainable for the long term, became of great importance for animal conservation [8]. It is known that an abrupt population reduction can lead to heterozygosity losses and increased inbreeding. These factors increase the susceptibility to environmental changes and, consequently, to a greater threat of extinction. Furthermore, the decrease in genetic variability can reduce reproductive fitness, and resistance to infectious and parasitic diseases [9].

Many Neotropical deer species have shown a declining trend, such as *Blastocerus dichotomus*, *Hippocamelus antisensis*, *Hippocamelus bisulcus*, *Mazama nana*, *Mazama bororo*, *Mazama bricenii*, *Mazama chunyi*, *Ozotoceros bezoarticus*, *Pudu mephistophiles*, and *Pudu puda*. As an example, the Huemul (*H. bisulcus*) was once abundant, but nowadays, it probably has less than 1% of its historical number of individuals [10]. The Marsh deer (*B. dichotomus*), on the other hand, had its original geographical distribution drastically reduced, losing approximately 65% of the areas in over four decades [3].

The reductions in Neotropical deer species’ distribution are estimated to range from 40% to 90% [3], which represents a conservation status considerably worse when compared to other mammals in the world. Globally, in mammals, there is a 25% rate of threatened species, and 15% are data-deficient [11]. Data on the Neotropical deer species shows that 53% are under some degree of threat, while 17.6% are data-deficient due to the lack of studies to confirm their real status [12]. An example of the deficiency in the available information refers to the evolution and taxonomy of the Cervidae family [5,13,14]. Another obstacle refers to the increased number of threatened species once the taxonomy has been updated, especially for the *Mazama* species review, where chromosomal polymorphism can be commonly found [15,16].

The deer stand out for having one of the highest rates of karyotype evolution among mammals, reflecting a marked chromosomal fragility [5,17]. In the *Mazama* genus, there is an interspecific karyotype variation that ranges from 2N = 32 to 70. Regarding intraspecific variations, extraordinary examples can be found, such as that of the *M. americana* species, in which the diploid number of chromosomes ranges from 42 to 53. This species was recently considered as a complex of cryptic species, because seven cytotypes have been found so far and the karyotype variants were correlated with the geographic location of individuals [18]. The mating of animals from different cytotypes resulted in infertile or sub-fertile progenies, indicating reproductive isolation and supporting the evidence that several species have not yet been described [19,20].

For wild population conservation, the best strategy would certainly be the maintenance of natural habitats that allow the natural biological processes, ecological interactions, and natural selection. However, the speed of occupation of these areas and the anthropic actions are so fast that these efforts are, at times, insufficient. Thus, assisted reproductive technology (ART) associated with germplasm banks are important tools for in situ and ex situ conservation programs [21]. The ART can be used to reduce genetic diversity losses in small populations, enabling the equal contribution of all individuals to the next generations [22]. The collection and cryopreservation of gametes and embryos can considerably reduce the need to capture free-living individuals or to transport captive animals between institutions [23]. It is indicated that the sample collection is carried out not only in threatened species but also for those that are not yet at risk of extinction, in an attempt to retain the genetic diversity that is generally lost in rare species [24].

Many specialists have repeatedly warned about species extinction, which is occurring at an alarming rate with no evidence of slowing within the next few decades [25]. In this context, there is a motivation to explore other horizons in the reproductive biology of wild species [8]. The ARTs that are widely known, such as artificial insemination, superovulation, and in vitro fertilization, are fundamental, but challenging to research or integrate into conservation programs. Thus, new methods and tools have been developed and may enable the use of wild species germplasm [8,26].

## 2. Reproductive Seasonality

The Cervidae family is characterized by its diversity and extraordinary ability to adapt to different environmental conditions. As part of this feature, within this family, differences in terms of geographical distribution, habitat, ecology, feeding, morphology, as well as their reproductive biology (reproductive behavior, anatomy, seasonality, and physiology) have been observed [27]. Deer species have adopted different reproductive strategies according to their environment to maximize the survival of their offspring and to ensure that fawning occurs during the period of optimum quality of forage [28]. It is important to state that we cannot generalize any species as having a typical deer pattern. While some species exhibit highly seasonal reproduction patterns in temperate climates, others are completely seasonal in tropical regions [29].

Despite the great diversity of species, there is still a lack of information, funding, and scientific attention on the reproductive biology of tropical and Neotropical deer [30]. In contrast, a few select deer species, particularly those that have high commercial or recreational value, have concentrated more than 95% of the efforts of research [29]. Although some ARTs for propagation and genetic management are used in commercial species, these technologies can rarely be directly transferable to those threatened to extinction. This is mainly due to the great diversity of reproductive patterns, where even in closely related species, marked differences are found [31,32]. Still, it is important to pay attention to the reliability of scientific information, since changes in the taxonomic classification revalidated some species that were considered synonymous [33]. For example, many studies on biological, reproductive, and behavioral data for *M. nemorivaga* were officially described as being from the Amazonian *M. gouazoubira* [34].

It is well-established that for species from temperate regions, reproductive patterns are highly seasonal, being guided mainly by the photoperiod [35]. In contrast, most tropical and Neotropical species have shown weaker seasonality or are completely aseasonal [34], being widely accepted that reproductive activity could not be photo-periodically dependent, but more influenced by local climatic factors such as annual rainfall patterns [35,36,37] or food supply [35,36]. These factors can result in some degree of reproductive synchrony between individuals (peak fawning), although conceptions and fawning can occur at any time of the year [28].

For the Southern pudu (*Pudu puda*), a Neotropical species, some authors have described that reproduction is partially controlled by the photoperiod, also responding to other environmental signals, such as variation food supply. Furthermore, this species appears to be more flexible in response to photoperiodic cues than other deer from temperate regions, since it exhibits postpartum estrus [38]. Studies have shown that, like the Northern pudu (*P. mephistophiles*) [39], the Southern Pudu appears to have two reproductive periods per year or to reproduce for a longer period than previously documented [40,41]. In addition to the two previously mentioned species, the only Neotropical deer females that seem to be seasonally polyestric are the North Andean deer (*H. antisensis*) and Huemul (*H. bisulcus*) [38,42].

Reproductive seasonality can also vary according to the location in which the individuals are found, with different fawning periods between populations within the same species. Near the equator or at low latitudes, the variation in environmental conditions throughout the year is lower than at higher latitudes. Thus, for populations of white-tailed deer (*O. virginianus*) that evolved in areas closer to the Equator, females are cycling at any time of the year, while in populations that evolved at higher latitudes, the breeding season is shortened and may last less than 70 days [43,44,45]. Although to a lesser extent, the pampas deer (*O. bezoarticus*) also exhibits reproductive variations according to its distribution, in which fawning is not strictly seasonal and occurs more frequently in certain months of the year [46,47].

Since the majority of the Neotropical deer can reproduce at any time of the year, it seems likely that females of the genera *Blastocerus*, *Ozotoceros*, and *Mazama* are continuous polyestric [47]. In brown brocket deer females (*M. gouazoubira*) kept in captivity, the occurrence of estrous cycles throughout the year has already been confirmed [48]. In captive males of the same species, annual reproductive monitoring also demonstrated the lack of seasonality, with no morphological (testis size and antler cycle), endocrine (testosterone and cortisol levels), or seminal (volume, concentration, motility, and sperm morphology) variations correlated with environmental variations (temperature, photoperiod, and rainfall) [49].

The lack of seasonality of most Neotropical species can facilitate the use of ARTs, since it increases the available fertile time of hinds and bucks, allowing a greater number of collections per year and a greater number of offspring from each individual. For species or populations that present reproductive seasonality patterns, it is possible to verify periods of low gamete quality that can affect the use of artificial insemination (AI) and other ARTs [29]. Seasonality is also important when dealing with the translocation of individuals, since seasonal tropical species maintained in temperate regions may present high neonatal mortality in winter due to the inability to adequately thermoregulate [36].

## 3. Germplasm Banks

Germplasm banks are tools that strongly assist the conservation of wild animals when used together with ARTs, presenting advantages that have been widely discussed [8,21]. They are constituted of genetic material, such as semen, oocytes, embryos, cells, tissues, among others, which are cryopreserved for later use. Its main usage is to circumvent problems imposed by captivity, where only a few individuals might be favored for reproduction. Furthermore, the use of cryopreserved genetic material may be the best alternative to minimize the effects generated by artificial selection, aiming to recover the original allele frequencies present in wildlife animals [46].

As there are many issues associated with the release of wild animals, the germplasm banks have gained even more attention, since the reintroduction of germplasm may present advantages to recover genetic diversity of populations, such as overcoming behavioral problems, assisting the animals to perform their ecological functions normally, and decreasing the risk of disease transmission [50,51,52].

Regarding behavioral problems, it is known that captivity can promote changes that compromise the individuals’ understanding of knowing whom their predators are, finding sexual partners, fighting for territory, and how to feed when they are reintroduced to natural environments [53]. Captive breeding can result in artificial selection and cause permanent changes in behavior, reducing the diversity of temperament characteristics [54]. The success of reintroduction can be compromised if breeding programs ignore this diversity and inadvertently lead captive populations to domestication [55,56]. When reintroducing genetic material, animals would have the opportunity to experience all maternal learning, maintaining the behavioral pattern of the species, allowing its survival and reproduction. Thus, genomic banks can maintain a large number of preserved individuals and, consequently, high genetic diversity without loss by selection, where the genetic profile of the generation remains unchanged.

For deer from the same species but presenting genetic distances within the populations, there is important information to take into account, such as location and molecular profile, among others. Mixing materials from genetically distant individuals can lead to outbreeding depression, defined as reduced F1 or F2 fitness and hybrid progeny [57]. Thus, subspecies or evolutionarily significant units (ESUs) must have their evolutionary heritage recognized and protected, maintaining the evolutionary potential of the species. For example, researchers suggested the existence of a subspecies of *P. puda* (continental—*P. p. puda* and Chiloé Island—*P. p. chiloensis*), which indicates that these populations should not be mated [54], as also occurs with pampas deer [58,59].

For the possibility of later use of cryopreserved material, good practices must be established in the handling and maintenance of samples in nitrogen cylinders, as well as their standard analysis. Semen samples should be evaluated after thawing for basic characteristics, such as motility, sperm morphology, and acrosomal membrane condition. From this evaluation, it would be possible to establish the sample’s fertilizing capacity and the best strategy for its use in the future, where better quality samples could be considered for artificial insemination (AI) procedures, and those with some deficiencies would be better employed in other techniques, such as in vitro fertilization (IVF) or even for intracytoplasmic sperm injection (ICSI) [59].

The collection and cryopreservation of somatic cells and tissues have gained more attention. These materials not only preserve the genetic diversity of populations but provide an excellent resource for biological research [60]. From it, the information could be obtained at a molecular level, such as the discovery of genes and proteins associated with specific reproductive characteristics that could optimize reproductive management [8]. For somatic cells and tissues, it would be equally interesting to establish standard testing protocols to evaluate the post-thaw quality characteristics, such as cell viability, plating efficiency, number of days to confluence, culture to at least the tenth passage, and normal chromosomal structure. With this information, it would be possible to assess whether the cells would have the potential to be applied in techniques that lead to the production of offspring, or if their use should be restricted to techniques related to molecular studies [61].

Despite the importance of these banks, there are still few institutions that have established the germplasm cryopreservation of Neotropical deer. The Deer Research and Conservation Center (NUPECCE), located at the São Paulo State University (UNESP) in Brazil, is one of the largest maintainers of biological materials of deer species, presenting a germplasm bank with samples from ten species of Neotropical deer, being composed of semen, embryos, and cell lines of more than 1000 individuals. The Santiago Metropolitan Zoo, in Chile, and the Buenos Aires Zoo, in Argentina, maintain germplasm banks (sperm and cell lines) from *P. puda*. At the FrozenZoo, which belongs to the San Diego Zoo, there are strains of some species of deer, and they conduct chromosomal analyses of each species [61,62]. It is important to state the comparative costs of maintaining a germplasm bank and a captive population. At this moment, NUPECCE maintains a breeding stock of approximately 50 animals belonging to eight species, with an estimated cost of US $3500.00 per month, while in the germplasm bank, where only the systematic replacement of liquid N2 is needed, more than 1000 animals are kept for US $500.00 per month. Thus, the use of these banks as a genetic reserve is the most viable long-term possibility.

## 4. Semen Collection and Cryopreservation

Considering the development of germplasm banks, the collection and preservation of the semen offer a viable option for the maintenance of the male genome in the long term, since this material is easier to obtain and cheaper to store than female gametes and embryos. However, the knowledge to carry out the collection and freezing of gametes is still very limited or non-existent for most wild species [63]. Studies on sperm characteristics have shown considerable differences in the viability, function, and success of fertilization, even among related species [52]. Semen collection in deer is carried out by three main methods: electroejaculation (EE), artificial vagina (AV), and post-mortem epididymal recovery [29]. The collection method must be carefully chosen because it can influence the quality of the ejaculate and the health and well-being of the animals [64,65].

The EE is a safer technique for the collectors, being the main method used to obtain ejaculate from deer. Additionally, the collection does not depend on prior training of the animals, and this allows the application in a large number of individuals, including those with behavioral deviations (aggressiveness, loss of libido, and inability to breed) or even in free-living animals. It can also be used in seasonal breeding animals outside of the normal breeding season or to attempt the collection from prepubertal males. This technique requires the use of chemical restraint that could compromise the health of the animals, therefore successive collections are not recommended. It is suggested to use a minimum interval of one week for performing EE in *Blastocerus*, *Ozotoceros,* and *Mazama* sp. [66]. The anatomical characteristics of each species can influence the choice and arrangement of the electrodes used during EE. There may also be differences in the response to electrical stimuli, and for this reason, such protocols require adaptations based on individual responses [67]. For EE in *Mazama*, *Ozotoceros,* and *Blastocerus*, animals are typically stimulated using 250 to 750 mA, administered in a three-second stimulation cycle followed by a three-second rest for a total of ten stimuli. After a one to two minute resting period, the stimulation cycle is repeated to obtain ejaculates. Previous reports have suggested attempting at least three stimulation cycles per session for successful semen collection [24]. The EE has been reported as an invasive method that affects several aspects of animal welfare [68]. Even under the effect of general anesthesia, in *O. bezoarticus,* the EE was responsible for the increase in heart rate, pulse rate, and concentration of enzymes (creatine kinase, aspartate aminotransferase, and alkaline phosphatase), as well as a decrease in body temperature [69].

The use of an artificial vagina (AV) is considered a good technique because the ejaculate presents a quality similar to coitus, reducing the animal’s stress and the need for constant collection [70]. The difficulty in this technique is to previously train the animals to become accustomed to the operator and handling [71]. In attempts at the seminal collection with AV in *M. americana*, it was found that males with low sexual interest were assaulted by females in more than 50% of the collection sessions. Thus, for the success of the technique, it is expected that the greater the libido of the males, the greater the probability of success in obtaining ejaculates [72]. Using females in estrus as a live mannequin, it was possible to perform the lateral deviation of the penis towards AV, and to obtain the seminal collection in individuals of the genus *Mazama* (Figure 1A). The success rates vary in the literature, from 75% to 100% in *M. gouazoubira* [72] and from 50% to 70% in *M. americana* [72,73]. It is also possible to use imprinted animals for seminal collections by AV, where the males perform the mounts directly on the operator’s knee (Figure 1B). This type of collection has already been described in *O. bezoarticus* [74]. In addition, it has also been observed in *M. gouazoubira* [75].

Considering the possibility of using genetic material from animals that died unexpectedly or that were euthanized for medical reasons, post-mortem sperm recovery, both from the vas deferens and from the tail of the epididymis, can be used. One of the main issues related to the success of this technique is the time interval between the death of the animal and the sperm collection. Although the epididymis can be stored at 4 °C for up to four days without the loss of full sperm viability, storage time should preferentially be less than 24 h [76]. In a report using individuals from *M. gouazoubira*, washing techniques in the tail of the epididymis, around 5 h after death, proved to be efficient for obtaining sperm, in which a volume of 3 mL was obtained, presenting motility of 80%, sperm vigor with a score of 3, sperm concentration of 1.62 × 10^9^ cells/mL, and 43% of abnormal spermatozoa. The ejaculate was diluted in a commercial medium (Botu-bov^®^—Botucatu, Brazil) and after thawing presented a motility of 30% and a sperm vigor with a score of 3 [77].

An interesting characteristic observed in *M. nemorivaga* is the reddish coloration produced by the ejaculate when the animals are submitted to EE [78], and when submitted to the collection by VA, it presents a pink tone [79]. No blood cells or hemoglobin were found in the seminal plasma samples to justify this characteristic, with the presence of clusters of a red pigment in the seminal plasma that was not bound to sperm [80].

Research on basic seminal characterization and formulation of specific diluents for Neotropical deer are still preliminary, and cryopreservation protocols are often adapted. Regarding the extenders, cryopreservation of semen from *M gouazoubira* was performed using tris-citric acid buffer with 2.25% or 20% egg yolk, which resulted in sperm motility of around 30% after thawing [72]. An increase in the efficiency of the diluent was reported by changing the amount of egg yolk by 10%, observing an increase in sperm motility [66]. For the *M. nana*, tris-citric acid buffer with 10% egg yolk was also used, resulting in post-thaw motility of 33.2%. The addition of vitamin E in this diluent enabled obtaining better post-thaw motility, resulting in 38.7% for sperm motility [80]. In *M. americana*, the use of tris-citric acid with 10% yolk, tes-tris with 20% yolk, and tes-tris with 20% yolk with the addition of Equex were tested, and obtained post-thaw sperm motility of 16.33%, 5.44%, and 24.66%, respectively [81]. Once they tested three diluents, the authors reported that one of the difficulties encountered was to achieve the minimum volume to conduct the experiment, which was possible in 83.3% of the collections performed by EE. In a study that sought to determine certain physicochemical and microscopic characteristics of semen of *M. americana*, the authors described pH and osmolarity of 6.90 ± 0.74 and 297.74 ± 19.10 mOsm/kg respectively, in the ejaculate [82]. These results assisted in choosing the diluents used by Alvarez and collaborators [73], together with substances that were beneficial for seminal cryopreservation of species of the *Mazama* genus (vitamin E and Equex) to enable the formulation of a medium for better results in the cryopreservation of semen. Thus, different penetrating cryoprotectants (glycerol, ethylene glycol, and dimethylformamide) were tested in tris-citric acid medium, with egg yolk 20% and adding vitamin E and Equex. The results for post-thaw motility in medium containing glycerol, ethylene, and glycol dimethylformamide were 55.31% ± 7.39%, 48.13% ± 2.39%, and 55.94% ± 2.77%, respectively. A significant improvement in membrane integrity was also observed, as reported in a previous study [81], that found around 20% to 40% of integral membranes depending on the medium used, while around 42.5% to 46.5% of membrane integrity was observed [82] (Figure 2). These authors obtained the ejaculate from AV, which in domestic ruminants, has resulted in spermatozoa with less DNA fragmentation, higher motility, higher acrosomal integrity, and active mitochondria when compared to the use of electroejaculation [83].

In collections performed during the reproductive season of *O. bezoarticus*, two commercial media were used for seminal cryopreservation, and Triladyl obtained better results than Andromed for sperm motility (around 20% vs. 10% respectively, after 2 h post-thaw) and membrane integrity (71.3 ± 3.8 vs. 64.1 ± 4.6, respectively) [84]. The use of a commercial thinner (Botu-Bov^®^, Botucatu, Brazil) was also tested for *M. gouazoubira,* where samples obtained from three free-living individuals were cryopreserved with this medium and showed an average of 56.7% post-thaw motility, 43% of intact acrosomes, 73.7% of membrane integrity, and 0.5% of DNA fragmentation [78].

In domestic species, the relationship between the quality of sperm cryopreservation and the presence of certain seminal plasma proteins has already been observed [85]. Although there are no databases related to the function of each protein for wild species, it is possible to compare it with the information available for domestic species [86]. Thus, for future applications in the formulation of diluents for species of the genus *Mazama*, the profile of the seminal plasma proteins described for *M. nemorivaga* [87] could assist in the development of a new cryopreservation medium through the identification of biomarkers.

## 5. Control of the Estrous Cycle and Artificial Insemination

Among the assisted reproduction technologies, AI is considered one of the most promising for the conservation of wild species. It favors the rapid dissemination of genetic material and minimizes the risks, challenges, and costs involved in translocating animals for reproductive purposes [65]. However, the success of this technique involves several factors, such as the understanding of the reproductive cycle of females and its manipulation, effectiveness to detect estrus, and seminal quality [88].

The estrus synchronization protocols allow controlling the length of the estrous cycle by administering progestins, injecting prostaglandins and their synthetic analogues (e.g., cloprostenol), or a combination of both. The use of cloprostenol has already been reported in *B. dichotomus* at a dose of 530 µg/female, showing that when used up to day six of the estrous cycle, cloprostenol did not trigger an effective luteolytic response. Thus, if there is no response in the first application, the suggested interval between applications for this species is 15 days. The animals showed behavioral estrus on average 58 h after the end of the treatment (ranging from 40 to 64 h). Ovulation and formation of the corpus luteum were verified by monitoring fecal progestins [89]. In *M. gouazoubira*, cloprostenol has also been tested using two injections ten days apart, resulting in 100% (*n* = 6) of hinds expressing behavioral estrus after 40 to 69 h of the last application [24]. Another study was carried out with the use of cloprostenol (265 µg) in *M. gouazoubira* within 11 days between applications, resulting in the estrus of 100% of the animals (*n* = 6). In this study, the period between the end of treatment and the onset of estrus was 40 to 69 h, the duration of estrus was from 18 to 60 h, and the diameters of corpus luteum were 3.21 ± 0.19 mm [90].

As for the progestogens, sponges containing 50 mg of medroxyprogesterone in *M. gouazoubira* were used for 14 days (being changed on the seventh day), resulting in the synchronization of 100% of the animals (*n* = 6) [24]. Although the use of progestogens for 11 to 14 days has been frequent in deer, exposure to progesterone for prolonged periods ends up negatively affecting the antral follicles, reducing fertility rates [91]. Finally, the use of CIDR for 8 days was tested in conjunction with the application of cloprostenol upon removal of the device, resulting in 100% of hinds (*n* = 6) expressing behavioral estrus. The period between the end of treatment and the onset of estrus varied from 52 to 88 h, the duration of estrus varied from 24 to 52 h, and the diameters of corpus luteum were 4.85 ± 0.74 mm [92]. This protocol was later used to manipulate the estrous cycle in other species, such as *M. nana* [93].

Considering the use of less invasive hormonal protocols, oral progestogens were tested for estrus synchronization in Neotropical deer and proved to be a cheaper and less risky alternative for these species. Melengestrol acetate (MGA) can be administered to individuals with food at a dose of 1.0 mg/animal/day and is recommended to be divided into two periods of the day (morning and afternoon). The MGA-based protocol was applied in *M. americana*, *M. nana*, and *M. nemorivaga*, being used in a reproductive management routine for natural mating or artificial insemination procedures [94,95].

For the identification of estrus, the use of intact males is a strategy that does not bring major complications in smaller species, where the male is removed from the presence of the females after the mating permission is verified, not allowing the copulation. However, in larger animals, this approach could result in aggression against the handlers. Among the signals emitted by the males when the female is in the receptive period is the Flehmen reflex, licking of the vulva and urine, vocalization, resting of chin, and chasing the hind. Although some authors point out that in the absence of a male, hinds express poor or no behavioral estrus [26], in Neotropical species, such behaviors have already been described, and their occurrence is possible even in reactive females. Estrus signals can be visualized by behavioral and physical indicators. Females in estrus manifest an increase in the rate of urination, decreased reactivity to the presence of the handlers, lordosis response to the handlers, which is often accompanied by a rapid wagging of the tail (Figure 3A), vulvar hyperemia, and abundant mucoid translucent discharge. Studies have shown the occurrence of such signs in *B. dichotomus* [89], *M. americana* [95], and *M. nemorivaga* [96].

There are several ways to perform AI, including intravaginal, intracervical, and intrauterine procedures. The success in using each of them varies considerably according to the species of deer [97]. Despite being relatively simple, low cost, and minimally invasive, intrauterine AI with cervical transposition may have its applicability limited by anatomical structures in some species. The deposition of intravaginal semen, on the other hand, requires a greater number of sperm for pregnancy to be achieved. The use of frozen semen with vaginal deposition was tested for *M. gouazoubira*, but it did not result in pregnancy [24]. In *O. virginianus*, pregnancy rates ranging between 50% and 100% have been reported after intravaginal/intracervical insemination with frozen semen [98].

Despite being a small species, transcervical AI with cervical traction was possible for a *M. nemorivaga* using refrigerated semen. Three previous attempts to perform AI in this species were carried out, but due to the inadequacy of the instruments, cervical traction was not possible. In these cases, semen was deposited at the back of the vagina and none of the attempts resulted in a pregnancy [99].

For most species of Neotropical deer, anatomical features such as the small opening of the cervix, its length, and a large number of cervical rings interfere in transcervical intrauterine semen deposition [24]. Thereby, laparoscopic intrauterine placement of spermatozoa is the most reliable technique for AI, and deposition can be made as close as possible to the site of fertilization (Figure 3B) [24]. However, this is a more invasive technique and involves a surgical procedure. For *M. gouazoubira*, this technique together with the use of cryopreserved semen resulted in pregnancy for 50% of the procedures (4/8). The AI was performed 12 to 14 h after the beginning of natural estrus [99], which seems to be suitable for this species and others belonging to the genus *Mazama*.

## 6. Multiple Ovulation and Embryo Transfer

The multiple ovulation and embryo transfer (MOET) is a very relevant technique when considering the need to cryopreserve female genetic material, which is a very complex task. It enables to increase the number of embryos and, consequently, enhances the number of offspring produced during the female life span. This technique allows embryonic transfers between individuals from the same species (intraspecific ET) or between different species (interspecific ET), which permits the implantation of embryos of threatened species in domestic or wild species that are at risk of extinction [59].

Multiple ovulation, known as superovulation (SOV), increases the number of antral follicles and ovulation to exploit the reproductive potential of females and avoid excessive handling of the animals [100]. The efficiency of SOV is one of the factors that will determine the success of embryo collection following uterine flushing. Although it has already been performed in several species of deer, a great variation in the response to hormonal protocols for SOV was observed. Poor responses in wild ungulates could also be explained by stress, which can be induced by intense manipulation during the procedures [101].

The treatments proposed for SOV in deer involve the use of progesterone devices associated with a single injection of equine chorionic gonadotropin (eCG) and/or multiple injections of follicle-stimulating hormone (FSH) that are administered a few days before or at the moment of progesterone withdrawal [59]. For *M. gouazoubira*, three treatments for SOV were tested, using two protocols based on equine Chorionic Gonadotrophin (eCG) and the other used FSH dissolved in polyvinylpyrrolidone, which is a synthetic organic polymer that allows the slow release of FSH and avoids excessive handling of animals. With a 600 IU dose of eCG, an average of 3.40 ± 0.68 visible corpus luteum (CL) was observed, while when using a 300 IU dose of eCG, 1.40 ± 0.24 CLs were visualized. Using 250 IU of FSH dissolved in polyvinylpyrrolidone, less than one CL per animal was observed (0.80 ± 0.49) [102]. Another protocol tested a 700 IU of eCG and obtained 6.0 ± 1.7 CLs, while the application of 130 mg of FSH divided into eight equal doses resulted in 2.0 ± 0.3 CLs [94]. Despite obtaining better SOV responses in treatments where higher doses of eCG were used, this hormone caused a premature regression of the CL, indicating possible poor follicle quality [103].

The embryo collection is carried out through uterine flushing using a catheter, which can be inserted via transcervical or by laparoscopy/laparotomy techniques. Although the collection by transcervical route has already been successful in *O. virginianus* [104], this technique is unsuitable for most Neotropical deer due to difficulties in transposing the catheter through the cervix. Surgical procedures were applied seven days after insemination/copulation, showing success in embryo collection [22]. Among the techniques, laparoscopy is less invasive and presents lower chances of adhesions in the reproductive tract, which could lead to reduced fertility and embryo collection. To date, this technique could not be successfully applied to Neotropical deer species. Using laparotomy, a greater collection efficiency in *M. gouazoubira* species was obtained, resulting in up to eight ova and embryos [24]. Studies aiming to collect embryos were also carried out in *B. dichotomus* to assist both in situ and ex situ programs (Figure 4). Among the females that underwent hormonal treatments, 11 demonstrated estrus and copulated and, posteriorly, underwent a laparotomy surgical procedure to flush the uterus and tubas. Embryos from five females were obtained, and only one female had viable embryos [104]. Two viable embryos were vitrified and stored in liquid nitrogen. Subsequently, these embryos were rewarmed and transferred into a free-living female belonging to a population that had been reintroduced, and from which it was known to need genetic reinforcement. This example illustrates the exchange of genetic material between captivity and free-living animals through the insertion of germplasm.

## 7. In Vitro Embryo Production

Among the available ARTs, in vitro embryo production (IVEP) is considered the most efficient for the propagation of small populations [59]. The IVEP technology has advantages over AI and MOET, such as total flexibility in pairing between males and females, the possibility of obtaining a greater number of embryos in vivo collection per donor female, and allows the cryopreservation of oocytes and embryos. The technique allows recovering genetic material from females in critical situations, such as reproductive pathologies (i.e., endometritis and tubal obstruction) as well as for animals that died [105].

Considering the difficulties to maintain the post-thaw seminal quality and develop adequate superovulation and embryo collection protocols, investing in IVEP studies could be interesting for Neotropical deer species. Since it requires fewer viable sperm for fertilization, the technique can maximize the use of male genetic material [106]. Differently from AI, the IVEP allows producing multiple descendants with a single straw. Additionally, the intracytoplasmic sperm injection (ICSI) allows the use of semen samples with low quality, immobility, and even spermatids [59].

Despite all the advantages, this technique is considered expensive and includes different steps: obtaining oocytes and in vitro maturation (IVM), in vitro capacitation of spermatozoa, in vitro fertilization (IVF), and in vitro development of embryos (IVD) [59]. The low success rates of IVEP are associated with failure in some of these steps, such as the ability to obtain good-quality oocytes, the lack of knowledge of the optimal conditions for the maturation of these gametes and embryos, and uncertainties about their subsequent transfer [107].

To obtain oocytes in captivity hinds, a successful approach is the use of exogenous hormone treatment (to stimulate multiple follicle development and maturation) in conjunction with the LOPU (laparoscopic ovum pick-up) technique for aspiration of oocytes [108]. Similar to what was observed in MOET, hormonal protocols for ovarian stimulation present better responses when follicular growth hormones (FSH and eCG) were applied at the moment of the follicular wave emergence. Regarding oocyte collection methods, LOPU is the method of choice for medium and small deer, since it promotes faster recovery and the possibility of repeating the technique in the same female, being considered less invasive than laparotomy [109,110]. For animals that die unexpectedly (in captivity or nature), slicing of ovaries is the most widely used technique [100].

The hormonal protocols tested in deer from temperate climates have shown great variation in follicular growth, varying from 3.21 to 15 [110]. For Neotropical species, ovarian stimulation focusing on IVEP has already been carried out in *M. gouazoubira* and *M. americana*. For both, a protocol based on multiple FSH applications was used (Figure 5). In this protocol, CIDR^®^ was used for eight days with an intramuscular application of estradiol benzoate (25 mg) and the administration of eight doses of FSH (total of 130 mg) every 12 h from the fourth day until the seventh day. For *M. gouazoubira*, ovarian stimulation was considered satisfactory, and the average number of follicles observed after surgery was 17.07 ± 9.12 [111]. In *M. americana*, females from purebred crosses (buck and hind from the same cytotype) showed good follicular growth response. In hybrid females (buck and hind from different cytotypes), two did not respond to the protocol and three others had well-developed follicles, but in smaller quantities than pure females. Although the number of antral follicles was not measured, the study reports the total of oocytes aspirated by laparotomy in one of the ovaries of each female (contralateral ovariectomy performed from histological evaluation). For pure females, the mean number of obtained oocytes was 8.67 ± 3.06, while for the three hybrid females that responded to the treatment, the average was 2.8 ± 5.6 per female [19].

For deer in general, the LOPU technique has enabled an oocyte recovery rate ranging from 46% to 71% [110,111,112]. For *M. gouazoubira*, the only Neotropical species where LOPU was performed, an oocyte recovery rate of 52.84% was obtained, which resulted in the collection of an average of 7.15 ± 3.72 oocytes/female per surgery. However, the repetition of surgical procedures in the same animal resulted in adhesions in the ovaries from the third surgery onward. It may be necessary to refine the pressure used by the suction pump since a high rate of naked oocytes was found [113].

After collection, oocytes are submitted to IVM, where they undergo several physical-chemical changes until nuclear, cytoplasmic, and molecular maturation is completed. In deer, IVM for 24 h resulted in 50% to 78% of oocytes at metaphase II [108,109]. In *O. virginianus*, the kinetics of the nuclear progression of the oocytes was observed in vitro, and oocytes in the germinal vesicle stage were observed from 0 to 3 h. The maturation of the oocytes of *M. gouazoubira* collected by LOPU and matured for 24 h resulted in 64.52% of oocytes in MII. In addition, it was found that 14.2% of the oocytes were immature, 4.84% were degenerate, and 16.13% presented spontaneous activation by parthenogenesis [111].

The determination of follicular fluid composition may be important to ascertain the microenvironment involved in follicle development and as an indication of the nutritional needs of the oocyte. In addition, selection of competent oocytes requires the identification of molecular markers present in the follicular fluid [112]. Thus, a study was performed to characterize the protein profile of ovarian follicular fluid in *M. gouazoubira*. Five adult females received an ovarian stimulation treatment, and the follicular fluid was collected by laparoscopy from small/medium (≤3.5 mm) and large (>3.5 mm) follicles. Concentrations of soluble proteins in follicular fluid were measured and proteins were analyzed by 1D SDS-PAGE followed by tryptic digestion and tandem mass spectrometry. The authors identified 13 major proteins, but with no significant difference between follicle size class (small/medium vs. large) [113].

## 8. The Potential of Somatic Cell Sources

The development of ARTs in Neotropical deer species has shown great challenges and, often, discouraging results. Considering the difficulties in working with wild animals and the low number of individuals available for research, the construction of a germplasm bank that is representative of the populations’ biodiversity becomes a difficult task if the materials are obtained exclusively by gametes and embryos. Since it is easier to obtain and cryopreserve tissues and cells, these biological materials are considered very promising and, as discussed in the topic “Germplasm banks”, represent a major cryopreserved source for Neotropical deer today.

Thus, the advantages of the collection and maintenance of cells and tissues are: (a) possibility of multiplication through in vitro culture (differently from what is possible with haploid cells), (b) the successive chemical/physical constraint of animals is not necessary (as can happen in many cases to obtain gametes and embryos), (c) facilitates obtaining material from free-living animals, as it does not depend on specific collection and cryopreservation protocols, (d) materials can be obtained from animals shortly after birth (in which there is still no production of gametes), senile (gametes with low quality), outside the breeding season, animals with reproductive problems or that were castrated, and animals that died, and (e) enables the formation of germplasm banks that effectively represent the genetic diversity of populations, where the entire genome of individuals is preserved, avoiding the problems arising from genetic drift.

Specifically, in reproductive biology, the use of these cells has provided several potential strategies, such as nuclear transfer and stem cell technologies, as will be discussed below. They can be isolated from a variety of sources, including organs, muscles, and skin. New strategies have also been carried out to obtain samples from non-invasive methodologies, such as cells contained in the pillar bulb or exfoliated cells present in the feces. For *B. dichotomus* and some species of the genus *Mazama*, hair traps have already been tested and presented good performance in the collection of material [114]. The collection of feces can be greatly facilitated by the use of scat detection dogs trained specifically to obtain this material in deer. The obtaining of fecal samples with this technique has already been widely used to evaluate fecal DNA [114,115,116]. Given the similarity in fecal morphology, as well as the relative taxonomic proximity, studies have aimed at obtaining fecal exfoliated cells from goats as a model for deer. The results demonstrated that viable cells could be obtained even 24 h after defecation in samples lying at room temperature or refrigerated [116]. Non-invasive methods could be used to obtain material from a large number of individuals without the need for capture.

### 8.1. Somatic Cell Nuclear Transfer (SCNT)

The ability of cytoplasmic machinery within the oocyte to reprogram differentiated somatic cell nuclei has provided potential strategies to generate embryos and living offspring, making this technique extremely interesting for endangered wild species. However, factors that limit the development and use of the technique are low efficiency, the need for a higher quantity of oocytes, and low access to recipient-compatible species [117]. In addition, complete reprogramming with the development of full-term individuals is achieved in a minority of cloned embryos, with a large proportion of them presenting epigenetic defects resulting from altered genome methylation patterns [118].

Regarding the difficulty in obtaining oocytes to use as cytoplasts, inter-species cloning (iSCNT) is an alternative [119]. The generation of animals through iSCNT also presents several obstacles, such as compatibility of genomic/mitochondrial DNA, embryonic genomic activation of the donor nucleus (karyoplasts) by the recipient oocyte, and the availability of receptors for the reconstructed embryos [120]. To increase the chances of compatibility between karyoplasts and cytoplasts, the use of oocytes of phylogenetically similar domestic species is recommended. The use of oocytes between different genera can cause problems in activating the zygotic genome, leading to the blocking of embryonic development [121]. This block seems to be related to the mitochondrial chimerism of the embryos (heteroplasmy), since it contains mitochondrial DNA from both the recipient oocyte and the donor cell of the nucleus, leading to altered mitochondrial respiration [122].

Among the initial steps for the development of cloning is the evaluation of the cell line to be used as a carioplast. Issues such as the number of cell passages and cryopreservation/thawing have been related to the success rates of SCNT [60]. In studies using *M. gouazoubira* fibroblasts, it was observed that as more passages in culture were performed, the greater the number of fibroblasts with fragmented DNA. Passages four and seven demonstrated good viability after being frozen/warmed, however, low quality of the tenth passage was reported, demonstrating that its use was not suitable for other biotechnologies, including iSCNT [123].

Among Neotropical deer species, iSCNT has already been reported in *M. gouazoubira* and *P. puda*. For *M. gouazoubira*, goat and cattle cytoplasts were used. The reconstructed deer/goat embryos were able to develop until the morula stage (12.5%), whereas, for the reconstructed deer/cattle embryos, 5.9% of them reached the blastocyst stage (Figure 6). To assess the influence of heteroplasmy in these embryos, the mitochondrial activity was measured through the levels of expression of some of its genes (ATP6, COX3, and ND5). A similar pattern of expression was observed between reconstructed goat/goat and deer/cattle embryos. For SCNT that used cattle cytoplasts (cattle/cattle and deer/cattle), the pattern of gene expression was similar to that of bovine embryos produced by IVEP for all the genes after expression normalization of the mitochondrial activity, which was performed by mitochondria using the 16S gene [124].

For *P. puda*, iSCNT was performed using fibroblast taken from the ear as karyoplasts and bovine oocytes as cytoplasts. It was possible to obtain embryos up to the blastocyst stage with normal morphology and karyotype. About the development timetable, two blastomeres stages were observed on the second day, 8 to 16 cells were observed on the third day, and blastocyst could be found on the fourth day, with hatch from the pellucid zone on the seventh day. Embryos reconstructed from fourth pass fibroblasts produced only structures up to the 8–16 cell phase (9.5%). Using sixth pass fibroblasts, better results were obtained, with 53% reaching the 8–16 cell phase and where two embryos (4%) reached the blastocyst stage [62].

An important problem for the success in obtaining living individuals by this technique is the availability of receptors for the embryos produced. The availability of females of endangered species is quite limited and the use of domestic recipient females may generate incompatibility with the genotype of embryos of wild species, resulting in impossibility in the development of embryos [125]. Some authors postulate that, since only the extra-embryonic trophoblast is involved in the interactions associated with maternal tissues, the construction of a chimeric embryo may allow the clone to develop successfully. The embryo would be constituted by trophoblast from the same species of the surrogated female and the inner cell mass from the wild species, thus allowing for better implantation in a domestic foster female [126].

### 8.2. Use of Stem Cells

Several potential strategies for the use of stem cells have been developed recently, which has led to an increased interest in germplasm banks that preserve such samples. These methodologies offer the opportunity to produce individuals from the reprogramming of terminally differentiated cells in pluripotent lines or the trans-differentiation of adult stem cells into gametes [61]. To choose the cell lines to be used in reproductive biotechniques, some important considerations are the ease of access to the tissue, ease of deriving and maintaining the lines in vitro, and the ability of these cells to reprogram/differentiate themselves [127]. Thus, it would be interesting for several types of cells to be stored and evaluate which ones are more prone to differentiation into germ cells, or even to reprogram into induced pluripotency stem cells (iPSC). The standardization related to obtaining, isolating, culturing in vitro, as well as characterizing these cells to use them is extremely important.

In *B. dichotomus*, three adult stem cell lines (antler, fat, and skin) were cultured and characterized and were shown to be promising sources to assist in constructing germplasm banks due to their plasticity, high rate of proliferation, as well as ease of obtaining. In all cell types, the multipotency of cells can be proven by the ability to differentiate into adipocytes, osteocytes, and chondrocytes. They also have rapid cell doubling times (25.96 h for antler, 32.24 h for fat, and 33.32 h for skin) and have expressed pluripotency markers such as *OCT4*, *SOX2*, *NANOG*, *REX1*, and *LIN28* [128].

Other interesting sources to be maintained in the germplasm banks are the gonad tissues and the spermatogonial stem cells (SSCs). The ovarian tissue could be transplanted to individuals of the same species or immunodeficient mice, and through stimulation with exogenous gonadotropins, it would be possible for the follicles to develop and be aspirated through the skin of the recipients. Testicular fragments can also be transplanted to obtain meiosis resumption by seminiferous tubules with consequent sperm production. Studies about histomorphometric evaluation in the testicles of *M. gouazoubira* have shown that testicular biopsy can be performed without the need for castration and without causing deleterious effects to the testis [129].

Concerning the use of SSCs, they consist of a population capable of self-renewal that can be isolated, cryopreserved, and transplanted into a host testicle [130]. The SSC techniques, originally developed in mice, have been documented in a variety of species [131]. Interesting information that allows the development of this technique is the fact that the testicles have natural barriers against the action of the immune system (hematotesticular barrier), which permits the insertion of cells from other animals/species without rejection by the host organism. However, the greater the phylogenetic distance between donor and recipient, the lower the chance of obtaining acceptable results [132].

In Neotropical deer, an innovative approach would be the use of hybrid *M. americana* individuals, that is, descendants of parents belonging to different cytotypes, such as SSC hosts. Salviano and collaborators [20] evaluated the existence of reproductive isolation in this species and found that, when cytotypes belonging to different evolutionary lineages were crossed, the hybrid fawns were proven to be azoospermic. Thus, these individuals could be used as a host without having to undergo radiation and there would be no doubt regarding the origin of sperm. However, as there is a “natural depletion” of the germ cells of the hybrid individual, it may be indicated that the SSC transplant is performed in sexually immature individuals, aiming to inhibit the occurrence of vacuolization in the seminiferous epithelium. These animals would be taxonomically appropriate to receive SSC from males of the genus *Mazama* or even from other species of Neotropical deer.

Another approach that has been cited for the conservation of wild species is reprogramming adult cells to obtain induced pluripotent stem cells (iPSCs). The iPSCs are very similar to embryonic stem cells (ESC) regarding their gene expression and DNA methylation [133], having the capacity for unlimited expansion and differentiation in multiple cell lines. One of the main applications of these cells in wild animals would be their differentiation into germ cells and gametes [134]. Some studies have already demonstrated that it is possible to recreate the entire gametogenesis, including male and female meiosis, in the in vitro environment, and to generate healthy offspring [135].

Similar to SCNT, the iPSC reprogramming efficiency varies according to the cell types used, being lower in terminally differentiated cells and faster and more efficient in less differentiated or undifferentiated cells [136]. The use of multipotent strains originating from the antler, fat, and skin of *B. dichotomus* has not made it possible to obtain iPSC strains so far, with resistance to input the reprogramming transcription factors into the interior of the cells being observed, and the presence of colonies with partial reprogramming that did not develop [137]. However, a preliminary study showed that when these same multipotent strains were directly subjected to trans-differentiation into PGCs, fat and skin responded well to the protocol, and showed morphological changes, expression of surface proteins, and transcripts consistent with that of PGCs. The expression profiles of the transcripts for strains differentiated from fat suggest a differentiation similar to the early stages of PGCs (pre-migratory and migratory cells) and those of the skin appear to be in a stage of late migratory cells and onset of meiosis. The antler was not shown to be an adequate source for direct differentiation into PGCs [138].

One of the most rigorous tests to assess whether iPSC strains have been adequately reprogrammed is embryo tetraploid complementation. In this technique, an embryo that is in the first cleavage (2 cells) undergoes electrofusion and then becomes a tetraploid embryo (4n). Upon reaching the blastocyst stage, the potential iPSC strains are inserted into the blastocele, and thus: (a) the cells originating from the tetraploid embryo cannot collaborate for the formation of fetal tissues since they are 4n, and (b) iPSC cells are responsible for the development of all tissues of the fetus but do not contribute to the formation of fetal attachments [139]. This would be a very interesting strategy to be applied in wild species, in which tetraploid (4n) embryos of domestic species would receive the iPSCs of the species of interest. The transfer of these chimera embryos to domestic recipients would be less likely to suffer rejection since the trophoblast is constituted by cells of the domestic species, releasing the species-specific fetal maternal recognition factors. The individuals generated would not be chimeric and their constitution would be made exclusively by iPSC cells. Unlike animals generated by iSCNT, they would not have heteroplasmy (Figure 7).

## 9. Conclusions

The development and application of ARTs in the conservation of Neotropical deer is still a challenge, and its use in the production of offspring is sporadic and limited to scientific research. The lack of well-established protocols for the cryopreservation can cause problems in the future application of these techniques, since it can lead to the storage of cells with low quality and, consequently, low potential for use. As with the majority of other species, there is less genetic contribution from females of Neotropical deer due to difficulties in harvesting and cryopreserving oocytes and embryos. In this context, the somatic cell bank would be a potential source of genetic diversity for threatened populations and could ease collection and storage. Modern biotechnologies based on the use of somatic cells and stem cells must contribute enormously to the use of different approaches for the preservation of deer species.

It is necessary to deepen the studies related to the basic aspects of the reproductive biology of these species and to expand the knowledge in this area. In addition, specialists in the reproduction of Neotropical deer are needed to increase the number of studies. It is essential to exalt the role of multi-disciplinarity within the construction of this knowledge, in addition, other challenges include the limited number of individuals available for research, their reactive behavior, susceptibility to stress, and in some cases, the lack of specialized facilities and knowledge to manage the animals safely.

## Figures and Tables

**Figure 1 animals-11-01961-f001:**
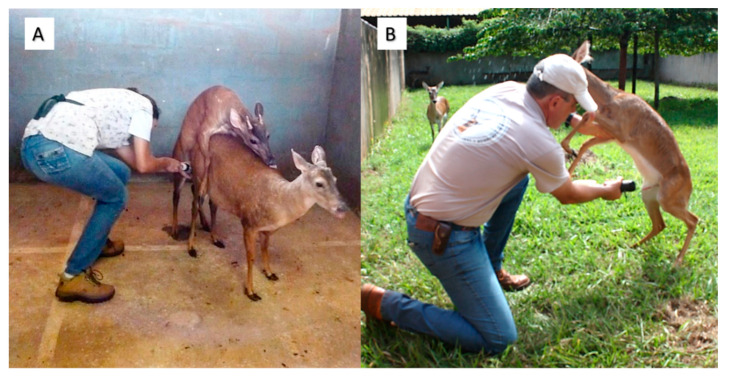
Seminal collection using an artificial vagina. (**A**) Use of female in estrus to perform lateral deviation of the penis. (**B**) Conducting collection in male with imprinting.

**Figure 2 animals-11-01961-f002:**
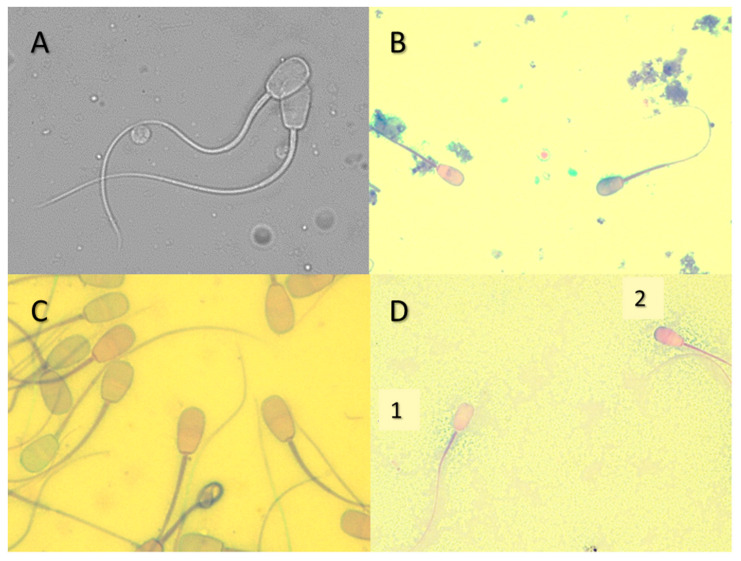
Spermatozoa from *M. americana*. (**A**) Spermatozoa in phase contrast to evaluate sperm morphology, (**B**–**D**) spermatozoa with simple acrosome dye to analyze their integrity, (**D1**) injured acrosome, and (**D2**) intact acrosome.

**Figure 3 animals-11-01961-f003:**
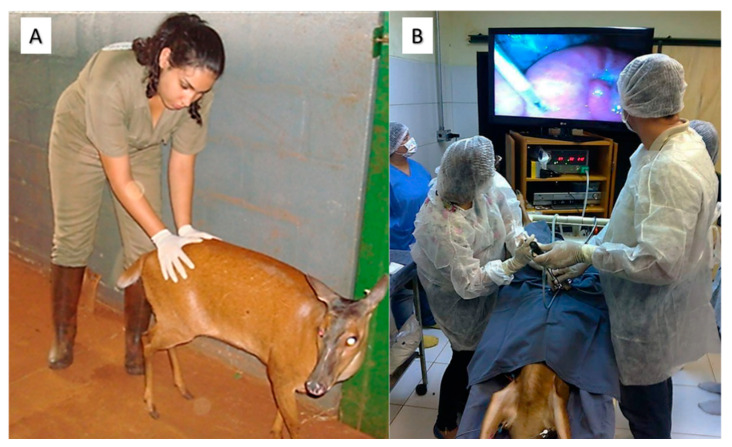
(**A**) *M. americana* female showing lordosis after having her posterior region manipulated by handler. (**B**) Artificial insemination by video-laparoscopy in *M. americana*.

**Figure 4 animals-11-01961-f004:**
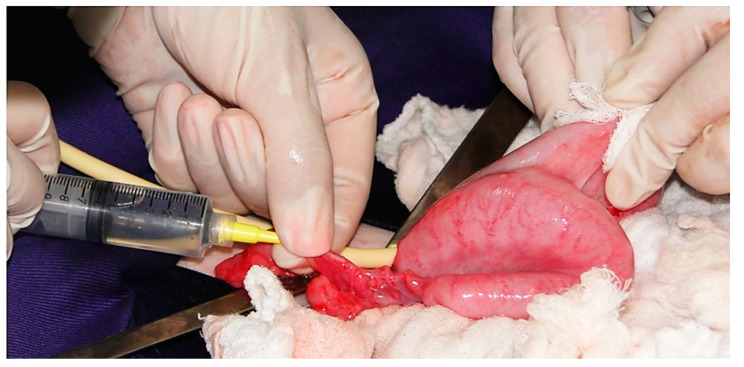
Uterine flushing for embryo collection in marsh deer (*B. dichotomus*).

**Figure 5 animals-11-01961-f005:**
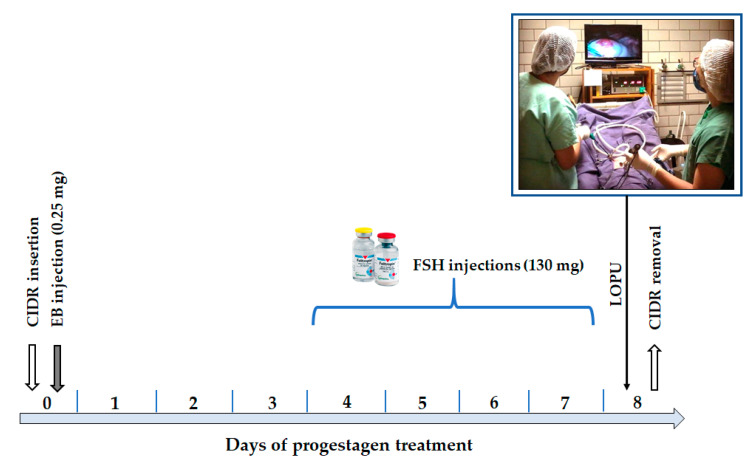
Schedule of the hormonal ovarian stimulation protocol used on females of *M. gouazoubira* and *M. americana*. CIDR: controlled internal drug release; EB: estradiol benzoate; FSH: follicle-stimulating hormone; LOPU: laparoscopic ovum pick-up.

**Figure 6 animals-11-01961-f006:**
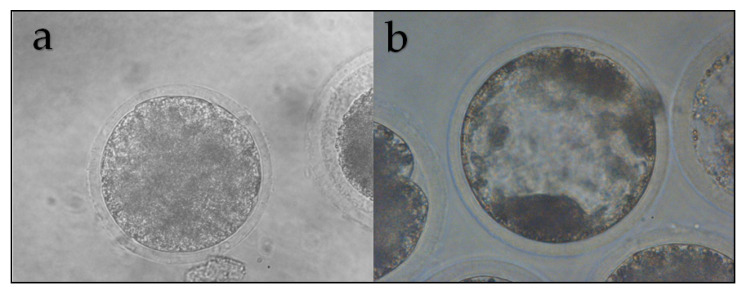
In vitro development of *M. gouazoubira* cloned embryos for the couples (fibroblast/cytoplast): deer/goat (**a**) and deer/cattle (**b**). 400×.

**Figure 7 animals-11-01961-f007:**
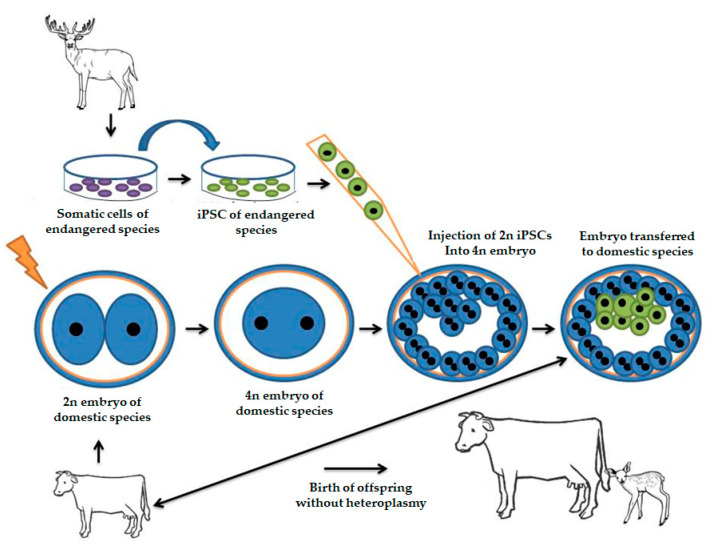
Scheme for production of individuals by the tetraploid complementation system. Somatic cells from wild species are reprogrammed into iPSC and inoculated in a tetraploid embryo of a domestic species. Tetraploid cells may give rise only to fetal attachments and iPSC cells to the fetus. Thus, there will be efficient maternal–fetal recognition after transferring the embryo to the recipient of the domestic species, and the offspring produced would not have the heteroplasm that is observed in the iSCNT.

## Data Availability

This review did not report any data.

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
