# Peer review of "Assisted Reproductive Technology in Neotropical Deer: A Model Approach to Preserving Genetic Diversity"

_animals, 2021, doi:10.3390/ani11071961_

Round 1
Reviewer 1 Report
Minor revisions
line 33: biotechnology instead of biotechnique
line 833: biotechnologies instead of biotechniques
Author Response
Dear Reviewer,
Dear reviewer,
First of all, we want to thank you for the suggestions and say that all have been accepted as described below and underlined in red in the manuscript.
Line 33: biotechnology instead of biotechnique
Answer: Thank for your suggestion. The word has been rewritten.
Line 833: biotechnologies instead of biotechniques
Answer: The word has been rewritten.
Reviewer 2 Report
The present manuscript aims to review the state of the ART in neotropical deer as a model approach to preserve the genetic diversity. The article is very interesting and it’s very well written. From the point of view of this reviewer, the manuscript is worth-publishing. Just small comments have arose.
L97-98. “In situ” and “ex situ” terms should be written in italics.
L167-171. The authors state that captive males from some specific deer breeds show seasonality with no changes in morphology, endocrine and seminal parameters. It makes no sense for this reviewer. How could the authors say that there is seasonality with no variations in the evaluated parameters? Could it be possible to get an explanation?
L412. Just as a curiosity. The dose for cloprostenol is 530 micrograms/female regardless the size of the female? Or it is 530 micrograma/kg?
L535. Same comment than line 97-98.
L552. Same comment. Write “in vivo” in italics.
L735, 739, 783. Same comment. Write “in vitro” in italics.
Author Response
Dear reviewer,
First of all, we want to thank you for the suggestions and say that all have been accepted as described below and underlined in yellow in the manuscript.
Line 97-98: “In situ” and “ex situ” terms should be written in italics.
Answer: Thank for your suggestion. The words have been changed.
Line 167-171: The authors state that captive males from some specific deer breeds show seasonality with no changes in morphology, endocrine and seminal parameters. It makes no sense for this reviewer. How could the authors say that there is seasonality with no variations in the evaluated parameters? Could it be possible to get an explanation?
Answer: We believe that the text was not clear enough, since the species mentioned does not present seasonality, and therefore does not present changes in morphology, endocrine and seminal parameters. We modified the sentence to “In captive males of the same species, annual reproductive monitoring also demonstrated the lack of seasonality, with no morphological…”
Line 412: Just as a curiosity. The dose for cloprostenol is 530 micrograms/female regardless the size of the female? Or it is 530 micrograms/kg?
Answer: Thank for your question. Yes, this work uses the same dose for all females, regardless of weight. So, we tried to be more specific by modifying the sentence by: a dose of 530 µg/female.
Line 535: Same comment than line 97-98.
Answer: The words has been changed.
Line 552: Same comment. Write “in vivo” in italics.
Answer: The words has been changed.
Lines 735, 739, 783: Same comment. Write “in vitro” in italics.
Answer: The words has been changed.
Reviewer 3 Report
L728: 8.1. Use of stem cells
L778: Please, explain abbreviation.
L783: in vitro in italics. Please, correct alongside the manuscript.
Author Response
Dear reviewer,
First of all, we want to thank you for the suggestions and say that all have been accepted as described below and underlined in blue in the manuscript.
L728: 8.1. Use of stem cells.
Answer: Thank for your suggestion. We changed for “8.2 Use of stem cells”, since the item “8.1” already exist.
L778: Please, explain abbreviation.
Answer: The abbreviation was provided (induced pluripotent stem cells – iPSC)
L783: in vitro in italics. Please, correct alongside the manuscript.
Answer: The words has been changed.